# A Review of Research on Mechanical Properties and Durability of Concrete Mixed with Wastewater from Ready-Mixed Concrete Plant

**DOI:** 10.3390/ma15041386

**Published:** 2022-02-13

**Authors:** Xianhua Yao, Junyi Xi, Junfeng Guan, Lijun Liu, Linjian Shangguan, Zhaowen Xu

**Affiliations:** 1School of Civil Engineering and Communication, North China University of Water Resources and Electric Power, Zhengzhou 450045, China; yaoxianhua@ncwu.edu.cn (X.Y.); x15565077384@126.com (J.X.); xvhewang@126.com (Z.X.); 2Zhengzhou Sanhe Hydraulic Machinery Co., Ltd., Zhengzhou 450000, China; liulijun2022@126.com; 3School of Technology Department, North China University of Water Resources and Electric Power, Zhengzhou 450045, China; sgljbh@163.com

**Keywords:** ready-mixed concrete plant wastewater, solid content, workability, mechanical properties, durability, microstructure

## Abstract

The wastewater from ready-mixed concrete plants is currently being recycled as concrete mixing water. It has attracted significant attention from the construction industry and researchers since it promotes sustainable development through environmental protection, energy-saving, and emissions reduction. This article review first introduces the nature of wastewater in ready-mixed concrete plants in different regions. Then the effects of solid content in water on various properties of concrete, including working performance, durability and microscopic properties, are reviewed, respectively, when concrete is mixed with wastewater instead of tap water. Furthermore, the microscopic mechanism of action in concrete mixing with wastewater is discussed, and future work is recommended. This review provides fundamentals on the study of the properties of concrete after wastewater is mixed into concrete.

## 1. Introduction

Persistently increasing demand for concrete and the advocacy of environmental protection, energy-saving, and greenhouse emissions reduction through waste concrete recycling has gained significant attention worldwide. Bu et al. [1] used recycled aggregate produced from construction waste to prepare recycled aggregate concrete, and Kelechi et al. [2] used waste tires ground into crumb rubber (CR) to replace fine aggregate in concrete to prepare self-compacting concrete. Ahmad et al. [3] used recycled concrete aggregate (RCA) as the coarse aggregate of concrete, while using waste glass (WG) as filler to prepare concrete. When the concrete batching plant produces concrete, a large amount of waste water will also be produced, and the wastewater can also be directly used in concrete production after treatment, thereby saving water resources. Therefore, waste recycling is becoming more and more popular. The primary wastewater sources in a ready-mixed concrete plant are washing aggregates, transport vehicles, mixing equipment, site cleaning and rainwater. In wastewater, the fine solid particles in wastewater are mainly cement hydration products, unhydrated cement particles, mineral admixtures (such as fly ash, silica fume, or metakaolin), and some ions from additives [4]. The wastewater is alkaline [5], and the random discharge of wastewater pollutes the environment and makes the water resources unable to be effectively utilized. Therefore, the direct use of wastewater from ready-mix concrete plants in concrete production is an efficient resource utilization method. It can help achieve the goal of zero net pollution and zero net emissions while also saving costs, thereby providing tremendous benefits. Therefore, experts and researchers have started taking a keen interest in the proposition. Kou et al. [6] treated the fresh concrete waste (FCW) produced by the cleaning equipment in the ready-mix concrete plant by combining precipitation, filtration, squeezing, dehydration, and air drying. Waste concrete was crushed into coarse aggregate, and used as a replacement for natural coarse granite in a certain proportion as the raw material of concrete production. Kou et al. [7] also pressed FCW into a fine aggregate (<5 mm), which replaced natural sand in concrete to produce partition wall bricks; Chen et al. [8] dried the wastewater mud by the filter press, and the dried powder was ground to be further used as a partial replacement of cement. However, several researchers have directly used it in concrete production by measuring the concentration of solid waste in wastewater and adjusting the water consumption according to the concentration [9]. The properties of concrete mixing water (such as solid content, admixtures and pH value) significantly influence the compressive strength, tensile strength and setting time of concrete [10,11]. When wastewater is used as concrete mixing water, the cement, fly ash, mineral powder, and other particles or wastewater additives will affect the solid content, and pH value of mixing water, thus affecting the resulting concrete performance.

Currently, regarding the use of wastewater from the ready-mix concrete plant as concrete mixing water, the concentration and dosage of wastewater are considered the main influencing factors. The change in the concentration and dosage of wastewater may be attributed to the change in total solid content in mixing water [12], and total dissolved solids (TDS) constitute the main impurities affecting the resulting concrete properties [13]. Therefore, in this paper, findings on the influence of solid content in wastewater on workability, mechanical properties, durability and microstructure of concrete with different strength grades, have been summarized.

## 2. Wastewater Properties

In order to save energy consumption, protect the environment and improve the level of comprehensive utilization of resources, the wastewater and slurry recycling system is established in the concrete ready-mix plant, and its main process flow chart is shown in Figure 1. Wastewater is mainly flushing production equipment, transportation equipment, pumping equipment, site flushing water and rainwater [14]. The sand and gravel separator is used to screen and wash the waste concrete generated after production is completed, and the separated sand and gravel are put back into the concrete production. The separated wastewater is used in the production of new concrete as production water after tertiary sedimentation in the sedimentation tank and the wastewater in the wastewater pool is generally a turbid liquid containing certain solid particles [15].

In different ready-mix concrete plants, the solid content of wastewater and the types of additives are different. The wastewater contains a large number of residual cement particles [9], which leads to an excessively high solid content and high alkalinity in the wastewater. The properties of wastewater will have a great influence on the workability, mechanical properties and durability of concrete. Regarding the research on the main components of wastewater, the researchers mainly analyzed the residue after drying of the wastewater, and the research results are shown in Table 1.

Chatveera et al. [21] magnified the microstructure of cement and wastewater powder by 10,000 times as shown in Figure 2. It was found that the cement particles were granular with different angles and were well hydrated, while the wastewater particles were interweaved by slender needle-like AFt and irregular flocculent C-S-H gel. Stamatis et al. [23] analyzed the particle distribution of concrete truck washing water sedimentation, as shown in Figure 3. It was found that wastewater particles with a particle size less than 10 μm account for 20%. When wastewater was used as concrete mixing water, the filling effect of wastewater particles can promote the improvement of concrete density.

In summary, the main components of wastewater are roughly the same in different regions and the main components of fresh wastewater are Ca(OH)_2_, SiO_2_, CaCO_3_, AFt, unhydrated C_2_S, C_3_S, and CaSO_4_•2H_2_O. The structure of fresh wastewater particles is relatively loose with a small particle size [16] of basically around 5 μm, and most of which are less than 10 μm [19]. It can act as a filler supplement and improve the particle gradation of the concrete system, and play a binding role. With the extension of storage time of wastewater, wastewater particles continue to hydrate [18], resulting in more C-S-H gel and AFt cemented with wastewater particles. The particle size of wastewater increases to basically around 10 μm, and mostly between 5 and 20 μm [19]. The activity of the wastewater particles keeps decreasing, and it can only be used as the filler of concrete pores until it loses its cohesive property. However, after the wastewater is stored for 10 days, and the unhydrated C_2_S and C_3_S can still be detected [18].

## 3. Influence of Solid Content in Mixed Water on the Performance of Concrete

The wastewater from the ready-mixed concrete plant is mixed with tap water and used as concrete mixing water. According to the slump and expansion of the mixture, the influence of the solid content in the mixed water on different strength grades is summarized in Table 2, and the influence on concrete with different water-cement (*w/c*) ratios is summarized in Table 3.

As shown in Table 2 and Table 3, the slump of concrete changed to varying degrees when wastewater was replaced with tap water in a certain proportion to prepare concrete. However, when the solid content of the mixed water is less than or equal to 6%, the change range of concrete slump is usually less than or equal to 30 mm. It shows that the incorporation of wastewater has little effect on the fluidity of concrete and, in most cases, the slump of concrete is reduced after the addition of wastewater. When waste water replaces tap water to mix concrete, the increase in the solid content of the mixing water leads to a decrease in the actual *w/c* ratio of the concrete, and the lack of cement hydration water, so the fluidity of the concrete decreases [29,30]. There are also admixtures such as retarders and water reducers in wastewater, which slow down the hydration process of cement and reduce the slump loss of concrete to a certain extent [16,20].

## 4. Influence of Solid Content in Mixed Water on the Compressive Strength of Concrete

The solid content in the mixed water has different effects on the compressive strength of various concretes. According to the 7-day and 28-day compressive strength of concrete, the influence of solid content in mixed water on different strength grades is summarized in Table 4, and the influence of different *w/c* ratios on concrete is summarized in Table 5. Meanwhile, the optimum solid content of mixed water used as concrete mixing water is summarized.

As is shown in Table 4, the compressive strengths of C20, C30, and C60 concretes mixed with mixed water are in the range of 90–115% of normal concrete. Under the condition of the same strength grade of concrete, with the increase in solid content in mixed water, most of its compressive strength shows the trend of first increasing and then decreasing, then increasing and then decreasing, as shown in Figure 4. There is an optimal replacement rate for different strength grades of concrete, so there is an optimal solid content in the mixed water. The optimum solid content in mixed water is mostly less than or equal to 5%. When the solid content in mixed water is at the optimum solid content, the compressive strength of concrete prepared by mixed water is higher than that prepared by tap water, regardless of whether the concrete is in the early or late curing stage.

In the research of C30 concrete, the optimal solid content of mixed water in references [4] and [18] is more than 5%. Qu [4] replaced fly ash particles with the solid content and other mass in the wastewater when mixing concrete, which led to differences between his research results and other researchers. Lu [18] found that the optimal solid content is higher, which may be related to the storage time of wastewater. Fresh wastewater is used when mixing concrete. The filling and binding effect of fresh wastewater particles is stronger, thus the optimal solid content of the mixed water was higher. In the research of C60 concrete, the optimal solid content in reference [15,24,30] is less than 1%, which may be due to the low concentration of wastewater itself (this may be related to the cleaning system and raw materials of the ready-mix concrete plants, and the solid content of wastewater is generally in the range of 5~15% [16]). After the tap water is replaced by wastewater, the solid content in the mixed water is also lower, resulting in a smaller scope of actual research. Therefore, the best measured solid content is lower.

As is shown in Table 5, most researchers believe that the compressive strength of wastewater concrete is 85–100% of that of ordinary concrete.

The alkaline environment provided by wastewater is conducive to the hydration reaction of cement. At the same time, the filling and bonding of wastewater particles enhance the compactness of concrete and promote the improvement of compressive strength of concrete. When the tap water is replaced by wastewater too much, the alkalinity of the concrete matrix increases. As a result, the cohesion of the transition zone between cement paste and aggregate is weakened, and the compressive strength of concrete is reduced [21,27]. In addition, when the solid content in mixed water is too high, the pores in concrete are completely filled with waste water particles. Additional solid particles will have adverse physical and chemical effects on it, and weaken the beneficial effects on concrete strength [13].

The mechanical properties of concrete are closely related to the curing age of concrete. Generally, with the extension of the curing age, the mechanical indexes of various mechanical properties will increase accordingly. In order to fully understand the characteristics of the later strength growth of concrete mixed with waste water replacing tap water in a certain proportion, scholars have also studied its compressive strength under long-term conditions. Li [25] found that the solid content in the mixed water increased from 0% to 1.8%, 3%, 4.2%, and 6% in the process. During the 60-day curing period, the compressive strengths of C20 and C40 concretes mixed with water were slightly higher than those prepared with tap water. Chatveera et al. [21] found that the compressive strength of concrete mixed with tap water was between 85–94% of that of concrete mixed with tap water at a curing age of 90 days. When the solid content of the mixed water is less than 2.53%, the compressive strength of the concrete exceeds 90% of that of the tap water mixed concrete. Yang [15] found that when using wastewater instead of tap water to mix concrete, the solid content of the mixed water increased from 0% to 0.42%, 0.84%, and 1.26% in the process. Its 90-day compressive strength decreased by about 10%. Lu [18] used fresh wastewater to mix C30 concrete. As the solid content in the mixed water increased from 0% to 3%, 6%, 9%, 12% and 15%, its 90-day compressive strength was changed by 42 MPa and continued to increase to 45.9, 50.4, 57.6, 62.2 and 64.5 MPa, with a maximum increase of 53.6%. To sum up, using waste water instead of tap water to mix concrete has little influence on the compressive strength of concrete at a long age. Its compressive strength is above 90% of that of tap water mixed concrete.

According to the research results of different scholars, the ratios of the 28-day compressive strength of concrete mixed with water with different solid contents and concrete mixed with tap water were calculated, as shown in Figure 5, and the ratios are between y = 0.88 and y = 1.10, obviously. Therefore, according to the existing test data, it can be inferred that when the tap water is replaced by wastewater in a certain proportion, the values of the compressive strength of concrete for wastewater mixed concrete are 0.88–1.10 times that for the tap water mixed concrete.

In summary, domestic scholars (in China) have shown that the compressive strength of wastewater concrete is 95–115% of ordinary concrete, while the results of foreign researchers are generally lower than domestic ones, most of which are 85–100%. This may be caused by different wastewater treatment technologies, additives and mineral admixtures (such as fly ash and silica fume) at home and abroad.

## 5. Effect of Wastewater on the Durability of Concrete

The durability of concrete has long been a concern for people, and various substances contained in wastewater will also have an influence on concrete durability.

### 5.1. Chloride Ion Penetration Resistance

Most researchers used the electric flux method to study the anti-chloride ion permeability of concrete. Under the condition of the same strength grade of concrete, the chloride ion penetration resistance of concrete becomes stronger with the increase of the solid content [4] as the fine particles in the wastewater play their role as fillers, fill the pores of the concrete and improve the density of the concrete [20,25]. Qu [4] used wastewater to replace tap water in different proportions, and the solids in the wastewater replaced the fly ash particles, and then mixed C20, C30, C40, and C50 concrete. The test results are shown in Figure 6. For the concrete of the same strength grade, with the increase in the solid content in the mixed water, the electric flux of the concrete at 28 d gradually decreased, and the C20, C30, C40, and C50 concretes decreased by up to 26.2%, 21.9%, 15.4%, and 12.5%, respectively. Therefore, when the concrete strength grade does not exceed C30, the effect of wastewater on the resistance to chloride ion permeability of concrete is great, and its effect is more obvious with the increase in the dosage. Zhou [20] used high-concentration wastewater to replace tap water in different proportions to mix C30 and C60 concrete. The research found that with the increase in the wastewater replacement rate, the average value of the electric flux in the test basically showed a downward trend, and the average value of the electric flux in all the tests was less than 1000 C. Therefore, the increase in the replacement rate of wastewater is beneficial to improve the resistance to chloride ion penetration, and the improvement effect of wastewater on chloride ion penetration resistance of C30 concrete is higher than that of C60 concrete. Li [25] used wastewater to replace tap water in different proportions to mix C20, C40, C60, and C80 concrete. The research found that with the increase in the wastewater replacement rate, the average value of the electric flux in the test basically showed a downward trend, and the average value of the electric flux in all the tests was less than 1000 C. Xiang et al. [16] found that under the condition of the same wastewater replacement rate, the effect of wastewater on the resistance to chloride ion penetration of C60 concrete is greater than that of C40 concrete. However, Chatveera et al. [28] found that the permeability of concrete prepared with waste water was between 86% and 106% of that of concrete prepared with tap water. In the preparation of concrete with waste water, adding superplasticizer and fly ash would also make its permeability lower than that of the control concrete.

### 5.2. Carbonization Resistance

Researchers studied the effect of mixed water with different solid content on the anti-carbonization performance of different strength grades of concrete. As the solid content in the mixing water increases, the carbonation depth of concrete with different strength grades decreases [4,30], which indicates that the anti-carbonation performance of concrete can be improved by adding wastewater. Qu [4] used wastewater with a solid content of 10%, and replaced tap water with 10%, 20%, 30%, 40%, and 50% to mix C20, C30, C40, and C50 concrete. The data for 7 d, 14 d, and 28 d carbonization depth all showed a downward trend. With the increase in wastewater replacement rate, the presence of residual admixtures in wastewater improves the compactness of concrete and reduces the carbonation depth. Xiang et al. [16] found that the addition of wastewater improved the carbonation resistance of concrete, and the carbonation resistance of concrete with higher strength grades was significantly better than that of concrete with lower strength grades. The strong alkalinity of the wastewater causes the fly ash in the cementitious material to undergo secondary hydration reaction, filling the pores inside the concrete, thus improving the carbonization resistance of the concrete. Zhou [20] used high-concentration wastewater instead of tap water to mix concrete, and the carbonation depth of concrete after adding high-concentration wastewater was smaller than the benchmark. It shows that the addition of high-concentration wastewater can effectively improve the carbonation resistance of concrete. The main reason is that the addition of high-concentration wastewater greatly reduces the water-binder ratio, which significantly improves the density of concrete and effectively prevents the intrusion of carbon dioxide. Li [25] used wastewater to replace tap water in different proportions, mixed C20, C40, C60, and C80 concrete, and put the prepared test blocks into a carbonization box to accelerate carbonization. The test results are shown in Figure 7. The addition of wastewater can effectively improve the carbonation resistance of concrete, and even after adding wastewater, the carbonation depth values of C60 and C80 concrete at 3 d, 7 d, 14 d, 28 d and 60 d are all 0 mm. The strong alkalinity of the cement hydration products in the wastewater offsets the carbon dioxide dissolved in water, preventing the CO_2_ in the air from interacting with the hydration products of the cement stone in the concrete, thereby improving the carbonization resistance of the concrete, protecting the steel bars from corrosion, and improving the concrete structure.

### 5.3. Frost Resistance

Freeze-thaw cycle is the main environmental factor affecting the durability of concrete, and the mixing of wastewater has a certain influence on the frost resistance of concrete. Qu [4] used the slow freeze-thaw method to perform freeze-thaw tests on C20, C30, C40 and C50 concretes with solid content of 0%, 1%, 2%, and 3% in mixed water. The test results are shown in Figure 8. As the solid content of the mixed water increases, the mass loss rate and strength loss rate decrease slightly (Figure 8), however they are within the test error range. Lu [18] used a slow freezing method to freeze-thaw test C30 and C50 concrete with different wastewater content under the condition of the same storage time. With the increase in the solid content in the mixed water, the mass loss rate and strength loss rate gradually decrease compared with ordinary concrete. With the improvement of concrete strength grade, the reduction range of mass loss rate gradually decreases. Chang et al. [9] mixed C25, C30 and C35 concrete with tap water and 15% solid wastewater, respectively. After 251 freeze–thaw cycles, the strength loss rate and mass loss rate of concrete mixed with wastewater are lower than those mixed with tap water.

Therefore, it can be concluded that the ready-mixed concrete plant wastewater can enhance the frost resistance of concrete. With the increase in the higher solid content in mixing water, it is more helpful to improve the frost resistance of concrete.

### 5.4. Sulfate Corrosion Resistance

Sulfate corrosion resistance of concrete is also a very important part of concrete durability research. Chatveera et al. [21] studied the influence of tap water and mixed water with different solid content on the sulfate attack resistance of concrete. It is found that with the increase in solid content, the mass loss of concrete gradually increases, which leads to the decrease in sulfate attack resistance of concrete. Lu [18] studied the influence of fresh wastewater with different substitution rates (0%, 20%, 40%, 60%, 80%, 100%) on sulfate resistance of C30 and C50 concrete under a dry–wet cycle. After 60 times of dry and wet cycles, the corrosion resistance coefficient of C30 concrete increases continuously, with the maximum increase of 4.65%, and that of C50 concrete also increases continuously, with the maximum increase of 1.45%. It shows that C30 and C50 concrete can effectively enhance its sulfate resistance by adding paste water. Chatveera et al. [28] studied that under the condition of containing fly ash (S) or superplasticizer (F), concrete was mixed with wastewater (W) with a solid content of 5–6%, and the acid resistance test of the concrete mixed with tap water (CC) was carried out in 5% sulfuric acid solution. The result of quality loss of concrete is shown in Figure 9. The sulfate resistance of concrete mixed with wastewater without fly ash or water reducer is lower than that of concrete mixed with tap water. However, when wastewater is used to mix concrete containing fly ash or superplasticizer, its sulfate resistance is better than that of tap water.

In summary, there are few studies on the impact of the use of ready-mixed concrete plant wastewater as mixing water on concrete durability at present, especially the frost resistance and sulfate resistance of concrete. However, from the research status, the solid content of mixed water increases, which promotes the improvement of concrete compactness and is beneficial to the improvement of chloride ion penetration resistance, carbonation resistance, frost resistance and sulfate erosion resistance of concrete. There are great differences in the research results of durability between domestic and foreign researchers, which may be caused by the differences in wastewater treatment technology, additive types and mineral admixtures when preparing concrete. Compared with foreign research, domestic researchers generally add fly ash and mineral powder to concrete. It carries out secondary hydration in the concrete, which weakens the adverse effect of the addition of wastewater on the concrete, thereby improving the durability of the concrete [31].

## 6. Effect of Solid Content on the Microscopic Properties of Concrete

The concentration of wastewater, dosage, standing time, and types of additives will all have an impact on the microstructure of concrete. He et al. [32] mixed C80 concrete mixed with A-type admixture (fly ash and mineral powder) or B-type admixture (fly ash and silica fume) with waste water and tap water, respectively, and then analyzed the X-ray diffraction (XRD) results of four kinds of cement pastes at 7 days and 28 days. The results show that the content of Ca(OH)_2_ in tap water and waste water mixed cement paste tested by XRD is similar at 7 days, and the content of Ca(OH)_2_ in tap water mixed cement paste at 28th day is slightly lower than the peak value of Ca(OH)_2_ in waste water mixed cement paste. Li [25] analyzed four kinds of cement pastes with a scanning electron microscope (SEM). The results show that the alkaline environment provided by wastewater is beneficial to hydration reaction and generates C-S-H gel at 7 days, which fills in the pores and makes the concrete more compact. At 28 days, the addition of wastewater reduces the porosity of the concrete and enhances the cohesion between the colloidal aggregates. Therefore, it promotes the improvement of the concrete’s compressive strength. Qu [4] mixed cement paste with tap water and wastewater, respectively, and tested its pore structure at different ages. The results of a mercury intrusion test are shown in Figure 10. Compared to the concrete mixed with tap water, the porosity of wastewater concrete is slightly reduced, and the most probable pore diameter is reduced by 12.4% and 8.3% at 1 d and 2 d, respectively, which is a large reduction. It shows that mixing concrete with waste paste enhances its compactness.

In summary, only a few researchers have studied the changes of microstructure and properties of concrete mixed with wastewater in previous studies. The mixed wastewater will change the material composition and its content of concrete and, at the same time, also affect the microstructure of concrete. Macro-mechanical properties are the manifestation of microstructure, and the changing trend of macro-mechanical properties can be accurately grasped only by clarifying the micro-mechanism of wastewater concrete performance change. Therefore, it is essential to study the micro-structure of wastewater concrete, and analyze its mechanism through its micro-means to provide the basis for the research of macro-performance change of concrete.

## 7. Conclusions and Prospects

A comprehensive literature review is presented herein on the performance of wastewater concrete. First, the properties of wastewater from a ready-mixed concrete plant are introduced. Then a great number of test results from different studies are collected, and focusing on comparing the influence of the solid content of wastewater on the work performances, mechanical properties, durability and microstructures of concrete. Finally, some suggestions are put forward for future work.

(1)The main components of wastewater in different regions are roughly the same, i.e., main cement hydration products and fine sands and gravel particles, however the solid content of wastewater is quite different. Therefore, if wastewater is recycled, the solid content of wastewater from the ready-mixed concrete plant should be detected simultaneously, and the solid content of wastewater should be adjusted as needed before use.(2)Concrete is prepared by replacing tap water with wastewater in a certain proportion. When the solid content of the mixed water is less than or equal to 6%, the change of concrete slump is within 30 mm. When the solid content in the mixed water is at the optimal solid content, the fluidity and compressive strength of the concrete change little compared to the concrete prepared from tap water, and the optimal solid content in the mixed water of wastewater is generally less than or equal to 5%.(3)The filling effect of wastewater particles can enhance the concrete durability to a certain extent. Currently, there are a few studies on the frost resistance and sulfate corrosion resistance of wastewater concrete. The influence of the solid content of wastewater on the frost resistance and sulfate resistance of concrete with different water-binder ratios is unclear, and as such, research on these aspects should be strengthened.(4)The storage time of wastewater greatly influences on the properties of wastewater particles, which leads to different effects of wastewater on various concrete properties. At present, there are limited studies on the influence of wastewater storage time on various properties of concrete. Further research should be carried out on the influence of wastewater storage time on concrete.(5)At present, there are few researches on the influence of wastewater on the microstructure of concrete. However, the changes of macroscopic mechanical properties of concrete are the external manifestation of the changes of microstructure. Therefore, we should strengthen the research in this area to provide reference for evaluating the changes of macroscopic mechanical properties of concrete.

## Figures and Tables

**Figure 1 materials-15-01386-f001:**
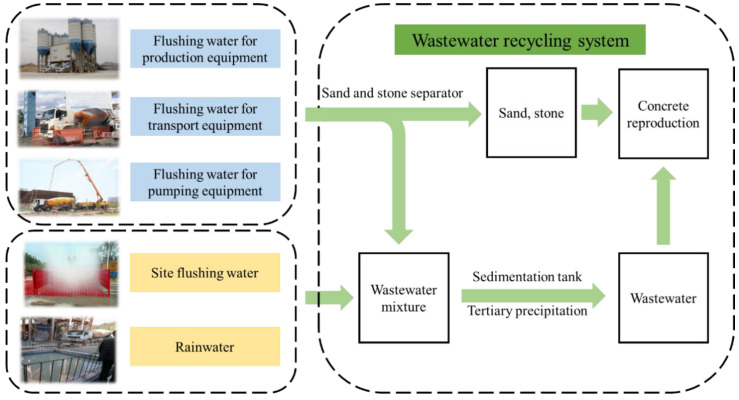
Main process flow of a wastewater recycling system.

**Figure 2 materials-15-01386-f002:**
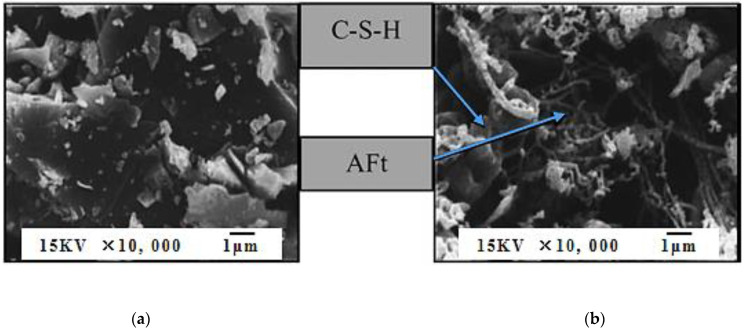
Micrograph of particles at 10,000-time magnification, (**a**) Portland cement Type I, (**b**) Wastewater powder. [21].

**Figure 3 materials-15-01386-f003:**
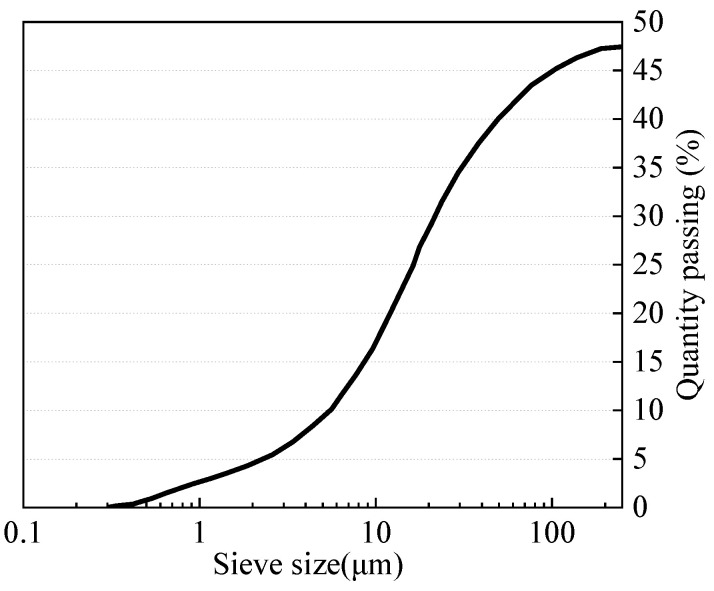
Particle size distribution of the number of wastewater particles [23].

**Figure 4 materials-15-01386-f004:**
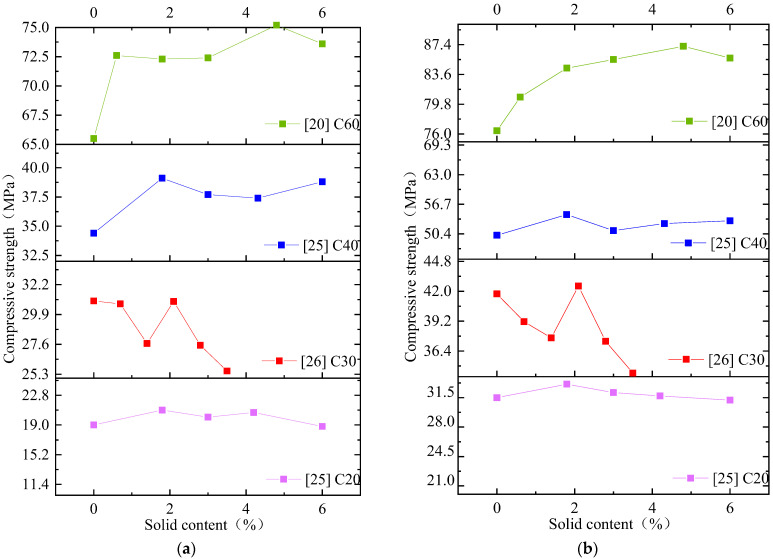
Variation diagram of compressive strength of concrete mixed with wastewater. (**a**) 7 days, (**b**) 28 days.

**Figure 5 materials-15-01386-f005:**
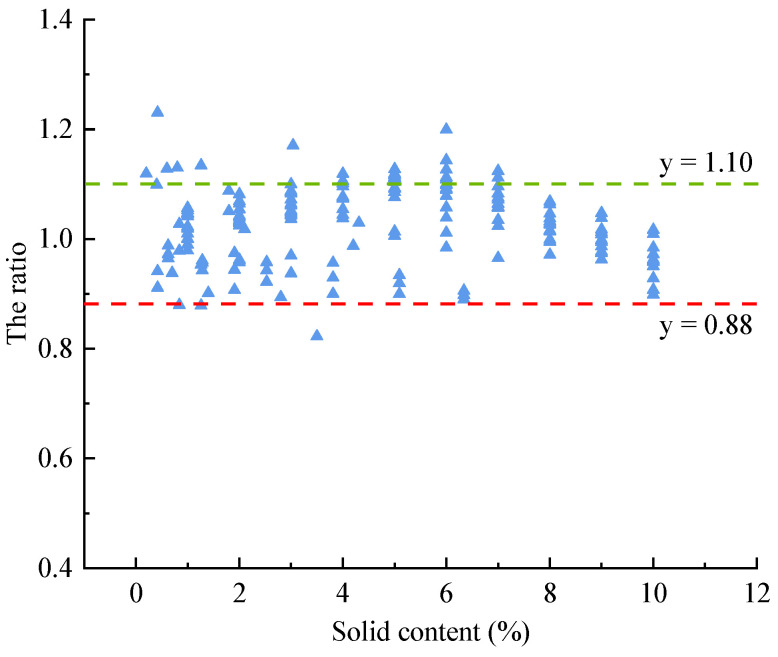
The relationship between compressive strength ratio and solid content.

**Figure 6 materials-15-01386-f006:**
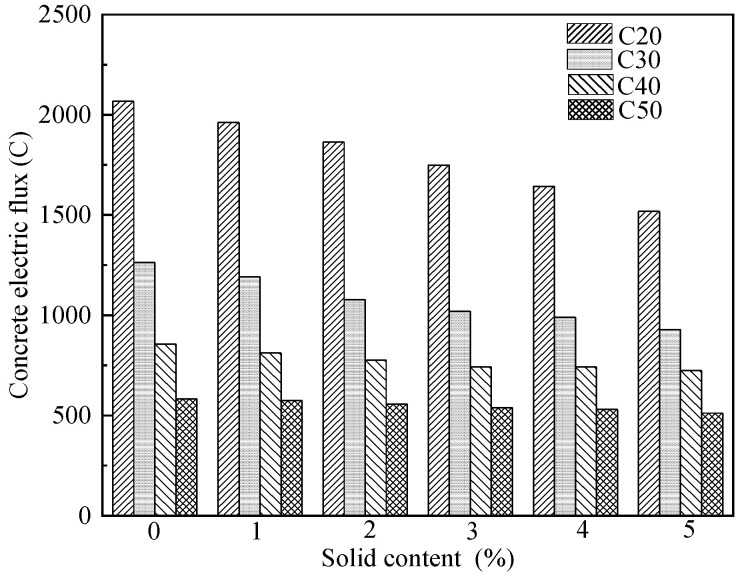
Effect of solid content on concrete electric flux [4].

**Figure 7 materials-15-01386-f007:**
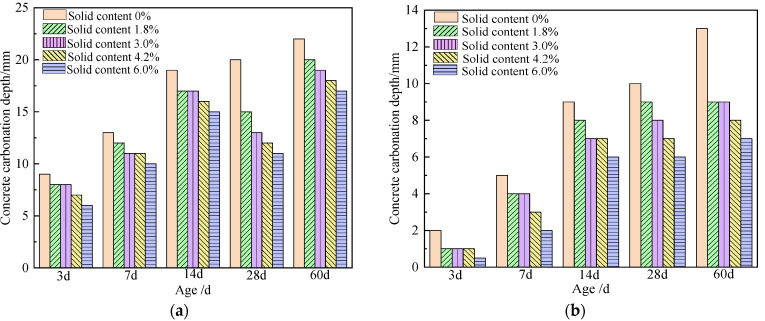
Effect of solid content on the depth of concrete carbonization [22]. (**a**) C20, (**b**) C40.

**Figure 8 materials-15-01386-f008:**
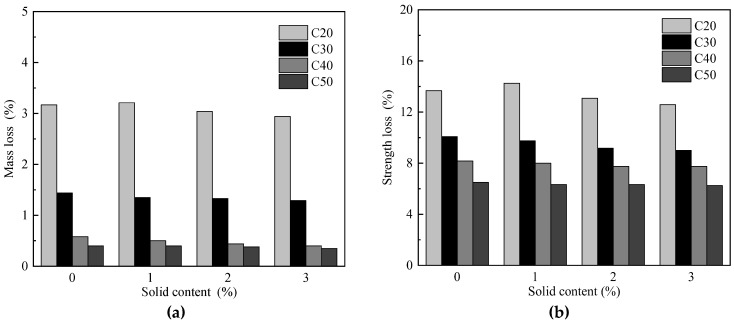
Effect of solid content on the frost resistance of concrete [4]. (**a**) Mass loss, (**b**) Strength loss.

**Figure 9 materials-15-01386-f009:**
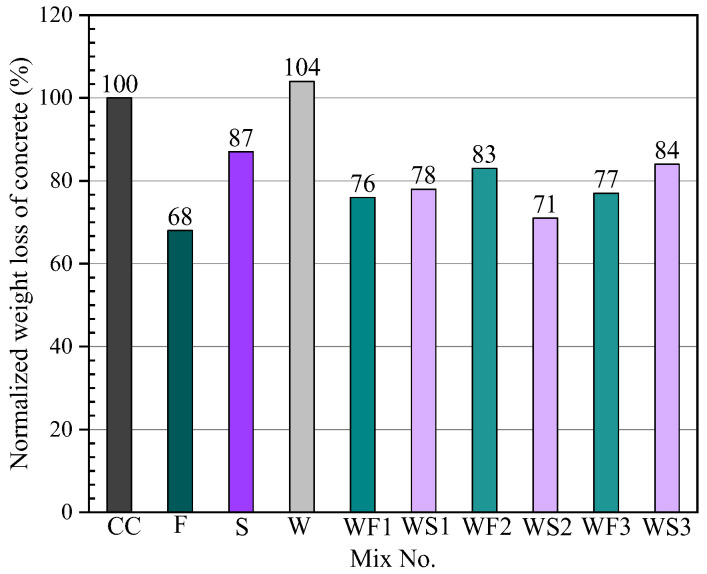
Acid corrosion resistance of concrete [28].

**Figure 10 materials-15-01386-f010:**
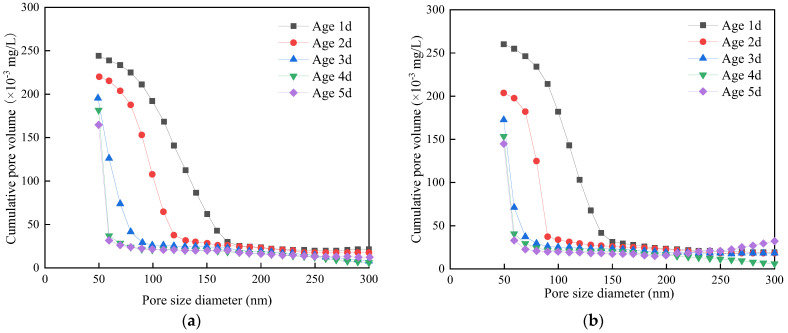
Pore diameter distribution of cement paste at different ages [4]. (**a**) mixed with tap water (**b**) mixed with wastewater.

**Table 1 materials-15-01386-t001:** Main components of wastewater solid particles.

Sources	Region	Test Substance	Main Components
[4]	Harbin, China	Cement paste mixed with wastewater	Ca(OH)_2_, AFt
[16]	Hunan, China	Drying wastewater sedimentation	Ca(OH)_2_, SiO_2_, CaCO_3_, CaSO_4_·2H_2_O
[17]	France	Sediment in wastewater sedimentation tank	SiO_2_, CaCO_3_, aggregates formed by Ca(OH)_2_ and/or C-S-H gel, C_2_S, C_3_S and CaSO_4_•2H_2_O
[18]	Sichuan, China	Wastewater	Ca(OH)_2_, AFt, C_2_S, C_3_S and CaSO_4_•2H_2_O
[19]	Tianjin, China	Fresh wastewater generated within 24 h	Containing Na-hydrated calcium sulfoaluminate (monosulfur type), containing Cl-hydrated calcium aluminateand and unhydrated C_2_S, C_3_S minerals, CaCO_3_ containing magnesium
		Store wastewater for more than 30 days	CaCO_3_, SiO_2_, AFt
[20]	Guangzhou, China	Wastewater	CaCO_3_, C-S-H, Ca(OH)_2_ and a small amount of mud powder
[21]	Thailand	Drying wastewater powder	Al_2_O_3_, Fe_2_O_3_, MgO, K_2_O, Na_2_O, SO_3_, and free CaO
[22]	Italy	Wastewater evaporation residue	CaCO_3_, SiO_2_
[23]	Greece	Wastewater sedimentation	CaO, Ca(OH)_2_, CaCO_3_, SiO_2_

Note: Ca(OH)_2_: Calcium hydroxide; AFt: Ettringite; SiO_2_: Silica; CaCO_3_: Calcium carbonate; CaSO_4_·2H_2_O: Gypsum; C-S-H: Hydrate calcium silicate; C_2_S(2CaO·SiO_2_): Dicalcium silicate; C_3_S(3CaO·SiO_2_): Tricalcium Silicate.

**Table 2 materials-15-01386-t002:** Influence of solid content on slump and expansion of concrete with different strength grades.

Concrete Strength Grade	Data Sources	Solid Content (%)	Slump Change Range(mm)
C20	[24]	0, 0.2, 0.4, 0.6, 0.8, 1.0	−20 ~ +20
	[25]	0, 1.8, 3.0, 4.2, 6.0	−25 ~ −5
C30	[4]	0, 1.0, 2.0, 3.0, 4.0, 5.0, 6.0	0 ~ +20
	[16]		−20 ~ −5
	[18]	0, 3.0, 6.0	−10 ~ 0
	[19]	0, 2.0, 5.0	−20 ~ −10
	[20]	0, 0.6, 1.8, 3.0, 4.8	−26 ~ +7
		0, 0.8, 2.4, 4.0, 6.4	−26 ~ +6
		0, 1.0, 3.0, 5.0	−28 ~ −6
		0, 1.2, 3.6, 6.0	−30 ~ −13
	[24]	0, 0.2, 0.4, 0.6, 0.8	−26 ~ −10
	[26]	0, 0.7, 1.4, 2.1, 2.8, 3.5	−10 ~ +10
C40	[19]	0, 2.0, 5.0	−30 ~ −20
	[24]	0, 0.2, 0.4, 0.6, 0.8, 1.0	−20 ~ −10
	[25]	0, 1.8, 3.0, 4.2, 6.0	−30 ~ −5
C50	[18]	0, 3.0, 6.0	−10 ~ 0
	[24]	0, 0.2, 0.4, 0.6, 0.8, 1.0	−10 ~ +10
C60	[4]	0, 1.0, 2.0, 3.0, 4.0, 5.0, 6.0	0 ~ +20
	[20]	0, 0.6, 1.8, 3.0, 4.8	−40 ~ −10
		0, 0.8, 2.4, 4.0	−44 ~ −5
	[24]	0, 0.2, 0.4, 0.6, 0.8, 1.0	−30 ~ 0

Note: 1. The strength grade of concrete refers to the compressive strength of concrete. It is expressed by the symbol C and the standard value of the compressive strength of the cube (in N/mm^2^; or MPa). 2. “−”indicates that slump or expansion degree decreases; “+” indicates that the slump or expansion degree increases.

**Table 3 materials-15-01386-t003:** Influence of solid content on slump of concrete with different *w/c*.

Data Sources	Solid Content (%)	*w/c*	Slump Change Range (mm)
[21]	0, 0.63, 1.27, 1.90, 2.54, 3.80, 5.07	0.5	−2 ~ −22
		0.6	−2 ~ −30
		0.7	−2 ~ −31
[22]	0, 0.82, 1.14, 2.56, 3.40, 3.99	0.57	−10 ~ −25
[23]	0, 0.13, 0.15	0.85	−10 ~ +90
[27]		0.45	+7 ~ +16
[28]	5.64	0.7	−10 ~ +5
		0.5	−5 ~ +5

Note: “−”indicates that slump or expansion degree decreases; “+” indicates that the slump or expansion degree increases.

**Table 4 materials-15-01386-t004:** Influence of solid content on the compressive strength of concrete with different strength grades.

Concrete Strength Grade	Data Sources	Solid Content (%)	The Ratio (%)		Optimum Solid Content (%)
			7 Days	28 Days	
C20	[24]	0, 0.2, 0.4, 0.6, 0.8, 1.0	103.8 ~ 112.1	94.5 ~ 106.1	0.8
	[25]	0, 1.8, 3.0, 4.2, 6.0	98.9 ~ 110.0	100.0 ~ 105.0	1.8
C30	[4]	0, 1.0, 2.0, 3.0, 4.0, 5.0, 6.0, 7.0, 8.0, 9.0, 10.0	124.4 ~ 147.4	111.0 ~ 116.0	6.5
	[9]	1.5	97.1	103.8	
	[15]	0, 0.42, 0.84, 1.26	93.5 ~ 100.1	88.3 ~ 91.1	
	[18]	0, 3.0, 6.0, 9.0, 12.0, 15.0	109.9 ~ 154.0	109.9 ~ 147.2	15
	[19]	0, 2.0, 5.0, 10.0	100.0 ~ 107.3	93.2 ~ 102.1	2.0 ~ 5.0
	[20]	0, 0.6, 1.8, 3.0, 4.8, 6.0	98.6 ~ 105.1	97.7 ~ 104.2	3.0
		0, 0.8, 2.4, 4.0, 6.4, 8.0	97.6 ~ 109.3	98.7 ~ 107.3	2.4
		0, 1.0, 3.0, 5.0, 8.0, 10.0	93.9 ~ 104.6	96.7 ~ 100.6	3.0
		0, 1.2, 3.6, 6.0, 9.6, 12	91.1 ~ 98.6	94.5 ~ 97.4	3.6
	[24]	0, 0.2, 0.4, 0.6, 0.8, 1.0	95.8 ~ 113.5	90.4 ~ 108.9	0.2
	[25]	0, 1.8, 3.0, 4.2, 6.0	108.7 ~ 113.6	102.0 ~ 108.7	1.8
	[26]	0, 0.7, 1.4, 2.1, 2.8, 3.5	82.5 ~ 100.0	82.8 ~ 102.6	2.10
C60	[4]	0, 1.0, 2.0, 3.0, 4.0, 5.0, 6.0, 7.0, 8.0, 9.0, 10.0	109.4 ~ 113.5	109.1 ~ 112.8	5.0
	[15]	0, 0.42, 0.84, 1.26	99.8 ~ 107.0	94.1 ~ 97.8	0.84
	[20]	0, 0.6, 1.8, 3.0, 4.8, 6.0	-	110.5 ~ 114.8	4.8
		0, 0.8, 2.4, 4.0, 6.4, 8.0	-	109.1 ~ 113.5	4.0
	[24]	0, 0.2, 0.4, 0.6, 0.8, 1.0	95.3 ~ 107.8	97.9 ~ 114.5	0.8
	[30]	0, 0.2, 0.4, 0.6, 0.8, 1.0	95.5 ~ 106.9	95.5 ~ 113.0	0.8

Note: “The ratio” indicates that the compressive strength of concrete mixed with mixed water and concrete mixed with tap water.

**Table 5 materials-15-01386-t005:** Influence of solid content on the compressive strength of concrete with different *w/c*.

Data Sources	Solid Content (%)	*w/c*	The Ratio (%)	
			7 Days	28 Days
[21]	0, 0.63, 1.27, 1.90, 2.54, 3.80, 5.07, 6.34	0.5	90.0 ~ 95.0	88.9 ~97.2
		0.6	88.9 ~ 95.2	89.8 ~ 96.4
		0.7	88.7 ~ 96.1	90.5 ~ 99.8
[22]	0, 0.82, 1.14, 2.56, 3.40, 3.99	0.57	100.0 ~ 103.0	95.9 ~ 100.1
[23]	0, 0.13, 0.15	0.85	96.0 ~ 106.7	94.9 ~ 106.2
[27]		0.45	80.9 ~ 97.5	89.3 ~ 100.5
[28]	5.64	0.7	77.0 ~ 84.7	76.6 ~ 86.2
		0.5	85.1 ~ 91.9	87.6 ~ 93.5

Note: “The ratio” indicates that the compressive strength of concrete mixed with mixed water and concrete mixed with tap water.

## Data Availability

All the relevant data and models used in the study have been provided in the form of figures and tables in the published article.

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
