# Peer review of "A Review of Research on Mechanical Properties and Durability of Concrete Mixed with Wastewater from Ready-Mixed Concrete Plant"

_materials, 2022, doi:10.3390/ma15041386_

Round 1
Reviewer 1 Report
Thank you very much for the opportunity to review the revised article. In my opinion, its scientific value has been significantly improved.
All the errors that I have highlighted are included in the correction. However, the article needs a little tweaking as noted below:
1) The author did not perform a statistical analysis of the measurement error. I don't know why? The graphs do not include the uncertainty of the result and the methodology does not describe how the uncertainty was calculated and what coefficients were selected for this. Please comment on this and make corrections.
2) Did the author check the influence of longer maturation time on the strength of the samples? Because the use of wastewater can slow down the hydration reaction in concrete?
Reviewer 2 Report
Manuscript is a review of existing research in the field of concrete mixed with wastewater. From this point of view (review article), the manuscript is well processed and contains interesting comparisons of results from previously published studies. Perhaps it would be more interesting to do a review more around the world (for example, in Table 1, most of the analyzed wastewater is from sources in China) but this is probably understandable, given the authors' country of origin. The review itself (Chapters 2 – 6) can be assessed as well done and interesting.
The only thing that should be improved is the conclusion. In Chapter 7 the conclusions need to be improved - some sentences are too general and must be improved to be more valuable: eg sentence “Different kinds of additives and mineral admixtures in wastewater influence the concrete workability and durability.” or “Macro-mechanical properties indicate microstructure changes in concrete.” is worthless (generally known). The conclusion (Chapter 7) must be improved and must contain a more valuable summary of the findings (comparisons) obtained from the review.
After incorporating the above comment (improving the quality of Chapter 7), I recommend the manuscript for publication.
Reviewer 3 Report
Comments to the Authors:
The authors of this paper present an interesting review, which summarizes findings on the influence of solid content in wastewater on work ability, mechanical properties, durability and microstructure of concrete with different strength grades. Nevertheless, some details should be considered by the authors:
COMMENT: The number of Introduction references is rather limited. More (recent) references could be added.
COMMENT: Page 4, Figure 2: The Micrographs presented in Figure 2 could be further commented.
They authors address the main questions posed, concerning the use of wastewater and the conclusions are consistent with the data, evidence and arguments presented. Thus, this article may be published.
Author Response
Please see the attachment.

This manuscript is a resubmission of an earlier submission. The following is a list of the peer review reports and author responses from that submission.
Round 1
Reviewer 1 Report
The literature research you present is of high importance, but it should be supported with your own material research. And then analyze.
The work is an introduction to research, it is a review of the literature. You should do your own research. As it stands, the work is not scientific enough.
technical fixes:
- instead of "slurry", use "paste" (ex. line 164.)
- "In summary, domestic scholars (in China) have shown..." (introduce it - line 169.)
Reviewer 2 Report
The abstract should get the essence of the paper and should be rewritten.
In general, the paper needs to be rewritten and more papers should be reviewed in support or against the topic in each section.
Avoid presenting many references for very well-known sentences. For example: In line 29: “The wastewater is alkaline [1-8],” just one reference is enough for this well-known sentence.
Moreover, more papers should be reviewed in support or against the hypothesis in each section. For example, in line 177: “Chloride ion penetration resistance” the influence of wastewater on the chloride resistance was superficially discussed. While the positive effect by Qu [1] was presented only, more data in support or against this data should be discussed.
The sentences are not well supported or reflected in the presented data. For example, Line 110: “When the solid content in mixed water is less than 5%, the slump of the concrete is reduced by less than 30mm.” This sentence cannot be directly inferred from Tables 4 &5.
In general, the paper is superficial and needs further attention and improvement.
Reviewer 3 Report
1) Please change the "Aft" to "AFt" line 2 table 1
2) The micrographs are very poor quality. Please add the new one or correct them. Where are the explanations for the arrows drawn? The arrows should also be of a different color than black.
3)Please remove the bold, line 2 table 4 and 5
4)It is very difficult to read the article if the authors provide the ranges of the achieved strength in%. Due to the very large number of variables, i.e. the constant content, it is difficult to precisely determine the result for the given values. Maybe the authors would try to present the tables in a graphic form?
5)Did the authors analyze the composition of wastewater, because the pollutants could to some extent support or delay the hydration process?
6)Have the authors analyzed the approach for geo-concrete? This is the extended maturation time of concrete and then determining its strength, e.g. 90 days?
7)Was the waste water pre-treated by filters or taken directly after the processes?
8)On the basis of so many research results, could it be tempted to determine the correlation coefficients for the decrease or increase in concrete strength, change in the rheology of the mixture or other parameters?
9) Were the results subjected to the statistical processing of the measurement error, because with such large discrepancies the confidence interval would be good for rejecting the results that do not meet the distorted measurement limits.